# Geographical Distribution of Dietary Patterns and Their Association with T2DM in Chinese Adults Aged 45 y and Above: A Nationwide Cross-Sectional Study

**DOI:** 10.3390/nu16010107

**Published:** 2023-12-28

**Authors:** Weihua Dong, Yuqian Li, Qingqing Man, Yu Zhang, Lianlong Yu, Rongping Zhao, Jian Zhang, Pengkun Song, Gangqiang Ding

**Affiliations:** 1National Institute for Nutrition and Health, Department of Geriatric and Clinical Nutrition, Chinese Center for Diseases Control and Prevention, Beijing 100050, China; dwh19861535785@163.com (W.D.); cnu_lyq@126.com (Y.L.); manqq@ninh.chinacdc.cn (Q.M.); zhangjian@ninh.chinacdc.cn (J.Z.); 2Key Laboratory of Trace Elements and Nutrition of National Health Commission, Beijing 100050, China; 3Chinese Center for Diseases Control and Prevention, Beijing 100050, China; zhangyu@ninh.chinacdc.cn; 4Shandong Center for Disease Control and Prevention, Jinan 250014, China; lianlong00a@163.com; 5Department of Clinical Nutrition, Sichuan Cancer Hospital and Institute, Sichuan Cancer Center, School of Medicine, University of Electronic Science and Technology of China, Chengdu 610056, China; zhaorongping7@163.com

**Keywords:** dietary patterns, T2DM, cross-sectional study, spatial statistical analysis

## Abstract

Background: This study aimed to investigate the geographical distribution of dietary patterns and their association with T2DM among Chinese adults aged 45 years and above. Methods: Data was from the China Adults Chronic Diseases and Nutrition Surveillance (2015). Dietary intake for each participant was determined through a combination of 3-day 24-h dietary recall interviews and food frequency questionnaires. Principal component analysis was used to extract dietary patterns and spatial analysis was employed to investigate the geographic distribution of them. T2DM was diagnosed using criteria of ADA 2018, and binary logistic regression was employed to examine the relationship between dietary patterns and T2DM. Results: A total of 36,648 participants were included in the study; 10.9% of them were diagnosed as T2DM. Three dietary patterns were identified with the name of plant-based pattern, animal-based pattern, and oriental traditional pattern, which were represented located in northern, northwest, and southern regions, respectively. After adjusting for potential confounders, participants in the highest quartile of the plant-based pattern were associated with lower T2DM odds (OR = 0.82, 95% CI: 0.74, 0.90) when comparing with the lowest quartile. However, participants inclined to higher quartiles of animal-based pattern had a higher risk of T2DM (OR = 1.15, 95% CI: 1.04, 1.27) compared with those in the lower quartiles. No significant association was found between the oriental traditional pattern and T2DM (OR = 1.03, 95% CI: 0.93, 1.14). Conclusion: Dietary patterns of Chinese population revealed geographical disparities, with plant-based dietary pattern showing protective effects and animal-based pattern carrying high risks for T2DM. Regional dietary variations and food environment are paramount in T2DM prevention and management.

## 1. Introduction

Type 2 diabetes mellitus (T2DM) is a significant public health issue and is recognized as one of the most prevalent chronic diseases worldwide. It poses a threat to human health and the global economy and is prioritized as a major health concern on the international health agenda [1]. Over the past few decades, T2DM has experienced exponential growth [2]. According to statistics reported by the International Diabetes Federation, there are currently 449 million people worldwide living with T2DM, and this number is projected to reach 702 million by 2045 [3]. In China, the prevalence of T2DM has risen significantly over the past 30 years. According to the latest data, the prevalence of diabetes among individuals aged 18 and above in China is 11.2%, with the number of diabetes patients expected to increase from 140.9 million in 2021 to 174.4 million in 2045 [4]. The majority of adult diabetes cases in China can be attributed to T2DM [5,6]. Considering the severe impact of T2DM on human health, it is crucial to conduct research on prevention strategies for this condition.

Dietary factors play a crucial role among the various preventive strategies for T2DM. Unhealthy diets elevate the risk of developing T2DM, whereas healthy diets such as low carbohydrate intake [7] and mediterranean dietary pattern [8] have demonstrated efficacy in enhancing blood sugar control and medication effectiveness. However, China’s intricate dietary culture necessitates a comprehensive examination of dietary patterns, as focusing solely on individual foods or nutrients might provide an incomplete understanding of the diet–disease relationship. Therefore, it is imperative to explore dietary patterns holistically. Notably, with profound changes in China’s development in recent decades, including shifts in dietary beliefs, food choices, nutritional status, and the prevalence of chronic diseases [9], studying the relationship between dietary patterns and T2DM has become increasingly important, as it can not only assess the potential of diet in preventing and managing T2DM but also significantly advance primary prevention efforts in T2DM control.

However, prior research has encountered limitations in elucidating the spatial dispersion of dietary patterns, which are notably influenced by distinct geographical contexts. Understanding the relationship between dietary patterns and geographical settings assumes pivotal significance in the endeavor to combat T2DM. Furthermore, age has been firmly established as a noteworthy T2DM risk factor, particularly within the middle-aged and elderly demographics (aged 45 years and above), considered as high-risk groups for T2DM [10,11,12]. Regrettably, there is currently a conspicuous dearth of research pertaining to the association between dietary patterns and T2DM in the Chinese populace aged 45 and above. In light of this, this study conducted a cross-sectional study based on the China Adults Chronic Diseases and Nutrition Surveillance (2015) for the following objectives: (1) investigate the principal constituents of dietary patterns in China; (2) examine the geographic distribution of dietary patterns in China; and (3) explore the association between dietary patterns and T2DM. The resulting insights of this study are poised to furnish us with substantiated dietary nutritional evidence, thereby catalyzing the advancement of elderly well-being and contributing to the management and prevention of T2DM.

## 2. Methods

### 2.1. Data Source and Survey Population

The data used in this study were from the China Adults Chronic Diseases and Nutrition Surveillance (2015) conducted throughout mainland China’s 31 provinces, autonomous regions, and municipalities (excluding Taiwan, Hong Kong, and Macao). Multi-stage stratified cluster random sampling method was applied to collect participants aged 18 years and above. The detailed description of the sampling procedure was described in reference [13]. This study was approved by the Ethics Committee of the Chinese Center for Disease Control and Prevention (Approval No. 201519-B), and informed consent was obtained from all participants prior to their enrollment.

Comprehensive information on participants’ basic demographics, lifestyles, diets, and health statuses was collected through household and personal questionnaires, body measurements, dietary surveys, and laboratory tests. This study specifically focused on participants aged 45 years or above. Those with missing values of basic demographic information were excluded from the final analysis, as were samples with missing or illogical values. Ultimately, 36,648 individuals aged 45 years or above were involved in this study (Appendix A).

### 2.2. General Information Collection and Dietary Nutrients Assessment

A standardized questionnaire was used to collect general demographic information, lifestyle, health status, and physical activity data through face-to-face interviews conducted by trained investigators at participants’ residencies. The dietary survey comprised a weighted dietary record, 3-day 24-h dietary recalls, and a food frequency questionnaire (FFQ) administered via personal interviews.

The 3-day 24-h dietary recalls were used to collect detailed information regarding all food consumed by participants in their homes and outside over three consecutive days (two weekdays and one weekend day), including staple foods, non-staple foods, snacks, fruits, and beverages. Participants were also asked to provide information on food frequency and consumption amounts for given foods in the past 12 months or one month using the FFQ. All food intake, including breakfast, lunch, dinner, and snack times, was documented during the interview period. Furthermore, a food-weighting record was used to gather information on the consumption of edible oil and primary condiments such as salt, sauce, and other seasonings during the three-day period.

Anthropometric measurements, including height, weight, waist circumference, and blood pressure were collected using standard methods [14] and certified measuring instruments (TZG height altimeter, Wuxi, China, and Tanita HD-390 electronic weight scale, Dongguan, China) approved and designated by the National Project Team, with readings that were accurate to 0.1 cm and 0.1 kg, respectively. When height and weight measurements were obtained, participants were asked to remove their shoes and hats and don lightweight clothing.

Fasting plasma glucose (FPG) were measured enzymically using an automatic biochemical analyzer (Hitachi 7600, Tokyo, Japan). Glycated hemoglobin (HbA1c) was determined by high-performance liquid chromatography (HPLC) using Trinity Biotech, Premier Hb9210 (Dublin, Ireland).

### 2.3. Ascertainment of T2DM

The definition of T2DM was based on the American Diabetes Association diagnostic criteria (2018) [15]. After excluding type 1 diabetes, gestational diabetes, and other special types of diabetes at the same time, T2DM was defined as follows: (1) FPG level ≥ 7.0 mmol/L and/or HbA1c concentration ≥ 6.5%; (2) current use of insulin or oral hypoglycemic agents; (3) a positive response to the question: “have you ever been diagnosed with diabetes by a doctor?”; and (4) individuals managing blood glucose through measures such as diet and exercise.

### 2.4. Identification of Dietary Patterns

Individual food items from FFQ were aggregated into 20 predefined food groups based on similarity of type of food and nutrient composition (Appendix A). The daily intake of various nutrients among research participants is assessed using the 3-day 24-h dietary recalls and food-weighting record.

Dietary patterns were extracted through principal component analysis, yielding standardized factor scores for each pattern as dietary pattern scores. The Kaiser–Meyer–Olkin measure of sample adequacy and Bartlett’s test of sphericity were applied to assess the sample adequacy for factor analysis. Orthogonal rotation was then employed to maximize variance and minimize cross-loading between factors, thereby enhancing the factor model’s explanatory power. The selection of patterns was guided by the use of scree plots, eigenvalues (>1), and the interpretability of each factor. Food groups with factor loadings ≥ 0.3 were identified as primary contributors to each pattern and were retained, with the cumulative variance explained by these factors capturing the variation in food intake. Each dietary pattern was labeled after the food category with the highest factor loading, mainly for descriptive interpretation. Subsequently, the patterns were grouped based on the quartiles of factor scores (Q1 to Q4) for further analysis. A higher score indicated one’s diet was closer to the corresponding dietary pattern.

### 2.5. Covariant Index

In this study, the covariates included age categorized into three subgroups (45–59 years, 60–74 years, and ≥75 years), educational level (below junior high school, junior high school, and high school or above), region (urban or rural), marital status (married or other), smoking (yes or no), alcohol consumption (yes or no), BMI (underweight BMI < 18.5 kg/m^2^, normal weight 18.5 ≤ BMI ≤ 23.9 kg/m^2^, overweight 24.0 ≤ BMI ≤ 27.9 kg/m^2^, and obese BMI ≥ 28.0 kg/m^2^), income (<5000 Yuan/month, 5000–9999 Yuan/month, ≥10,000 Yuan/month), a family history of chronic diseases (yes or no), and central obesity (yes or no). “Other” marital status includes unmarried, divorced, and widowed individuals. Smoking is defined as having smoked at least once in the past 30 days, while drinking is defined as having consumed alcohol at least once in the past year. Adequate physical activity is defined as engaging in moderate or above physical activity more than 150 min per typical week. BMI was calculated by dividing weight in kilograms by height in meters squared (kg/m^2^).

### 2.6. Statistical Analyze

Categorical variables were expressed as counts (percentages) and were compared using the chi-squared test. Continuous variables were expressed as mean (standard error, SE) if data presented a normal distribution and were compared using analysis of variance, or expressed as medians (25th, 75th), if data presented a skewed distribution and were compared using the Wilcoxon rank sum test. Pearson or Spearman correlation analysis was used to evaluate the correlation of food groups and nutrients with dietary patterns. After categorizing the dietary pattern scores into quartiles, a binary logistic regression model was used to examine the relationship between dietary patterns and T2DM, while adjusting for potential confounding covariates. The geographical distribution of dietary patterns was investigated through exploratory spatial dependency analysis of dietary pattern scores using global spatial autocorrelation and local spatial autocorrelation. The primary measure of spatial correlation, the global Moran’s I (GMI) statistic, is commonly employed. GMI values range from −1 to 1, where a GMI close to −1 indicates dispersed adherence to dietary pattern scores, while a GMI close to 1 suggests clustered adherence. A GMI of 0 indicates randomly distributed adherence. Statistically significant GMI values (*p* < 0.05) have the potential to reject the null hypothesis, signifying spatial autocorrelation. The outcomes of local spatial autocorrelation are portrayed through significant local clusters. High-high clusters and low-low clusters indicate areas with high or low scores surrounded by similar features, while low-high and high-low outlier illustrate areas with low scores surrounded by high scores, and vice versa. Hotspot analysis involves location-based weighting of variables, normalization of the resulting weighted values, and the construction of z-scores for hotspots (positive z indicating areas with higher values clustered closer together) or coldspots (negative z indicating areas with lower values clustered closer together), providing characteristics of spatially concentrated hotspots or coldspots. Dietary pattern scores by province were calculated and shown with choropleth maps. Statistical significance was defined as two-sided *p*-values less than 0.05. Statistical analyses were conducted using R 4.2.2 and SAS 9.4 (SAS Institute, Cary, NC, USA). Spatial analysis was performed using ArcGIS Desktop version 10.8 (ESRI, Inc., Redlands, CA, USA).

## 3. Results

### 3.1. Characteristics of Subjects according to T2DM Status

A total of 36,648 middle-aged participants (45 years and above) were included in this study from the 2015 China Adults Chronic Diseases and Nutrition Surveillance, among whom 3983 individuals had T2DM (10.9%). In this study, 19,982 (54.5%) were female, 21,988 (60.0%) were from rural areas, and 34,619 (94.5%) were married. The median age of all participants was 58.6 years. Individuals with T2DM had higher levels of BMI (25.7 [23.5; 28.1] vs. 23.9 [21.7; 26.3], *p* < 0.001), economic income (5.6% vs. 4.6%, *p* < 0.001), and educational attainment (5.8% vs. 3.4%, *p* < 0.001) compared to those without T2DM (Table 1).

### 3.2. Dietary Patterns

The factor loading matrix for dietary patterns is presented in Appendix A, with the inclusion of factor loadings with absolute values ≥ 0.30. Three common factors (dietary patterns) were identified, with eigenvalues of 2.54, 1.81, and 1.29, and a total common factor variance of 5.64, accounting for 28.18% of the variance in food group intake. These three dietary patterns were labeled as follows: the “Oriental traditional Pattern”, characterized by a higher intake of rice and products, pork, aquatic products, vegetables, poultry, and a lower intake of cereal foods; the “Animal-based Pattern”, characterized by a higher intake of processed meat products and offal, red meat, snacks, nuts, beverages, and fruits; and the “Plant-based Pattern”, characterized by higher intake of wheat and products, tubers, fungi and algae, legumes, fruits, vegetables, and eggs.

Figure 1 illustrates the characteristics of three dietary patterns, including the strength and significance of correlations between the three dietary patterns and food groups/nutrients (Figure 1a), and a comparison of food groups and nutrient intake in the most representative group (Q4) of each dietary pattern (Figure 1b). The plant-based pattern displayed the strongest correlation with dietary fiber (r  =  0.26; *p*  <  0.01), while the oriental traditional pattern exhibited strong correlations with MUFA, cholesterol, niacin, and vitamin A (r  > 0.30; *p*  <  0.01). The animal-based pattern demonstrated the strongest correlation with selenium (r  =  0.22; *p*  <  0.01) (Figure 1a).

Comparing the three dietary patterns, the plant-based pattern displayed significantly higher intake of dietary fiber, folate, and carbohydrates. Conversely, the oriental traditional pattern revealed significantly higher intake of energy, protein, fat, cholesterol, niacin, PUFA, MUFA, SFA, vitamin C, and β-carotene compared to the other two patterns. Furthermore, the animal-based pattern showed significantly higher intake of retinol relative to the other two patterns, with vitamin E intake being significantly higher than that of the oriental traditional pattern (Figure 1b).

### 3.3. Characteristics of Subjects according to Dietary Pattern Scores Quartile Distribution

Among the three dietary patterns, males tended to have higher scores. Participants with higher scores in the plant-based and oriental traditional patterns were typically younger, while those with higher scores in the animal-based pattern were much older. Individuals with higher scores in the plant-based and oriental traditional patterns were more likely to come from urban areas, whereas those with higher scores in the animal-based pattern were predominantly from rural areas. Participants with higher scores in the animal-based and plant-based patterns exhibited higher rates of smoking and alcohol consumption, whereas those with higher scores in the oriental traditional pattern had higher alcohol consumption rates and lower smoking rates.

Participants who scored higher in the oriental traditional pattern demonstrated elevated levels of BMI, FPG, and HbA1c. Furthermore, individuals with higher scores in the animal-based pattern exhibited higher FPG and HbA1c levels. Conversely, those with higher scores in the plant-based pattern displayed lower FPG level (Appendix A).

### 3.4. The Spatial Distribution of the Three Dietary Patterns

Figure 2(a-1–a-3) depict the actual distribution of three dietary patterns scores among Chinese adults aged 45 and above in 2015. The plant-based pattern was primarily concentrated in the northern region Figure 2(a-1), while the oriental traditional pattern was predominantly concentrated in the southern region of China Figure 2(a-2), and the animal-based pattern was mainly concentrated in the northwest region Figure 2(a-3).

Figure 2(b-1–b-3) illustrate the local spatial autocorrelation of three dietary patterns scores. The results of spatial autocorrelation analysis indicated a significant clustering trend among the three dietary patterns (Moran’s I: Plant-based Pattern = 0.433, Oriental traditional Pattern = 0.569, Animal-based Pattern = 0.529, *p* < 0.001). This suggested that areas with high scores tend to cluster together with neighboring areas also having high scores, while areas with low scores exhibit a similar clustering pattern.

Further local spatial analysis revealed that areas north of the Yellow River are highly clustered and hotspot regions for plant-based pattern, while areas south of the Yangtze River exhibit highly clustered and hotspot regions for oriental traditional pattern. High clustering and hotspots for the animal-based pattern, on the other hand, are concentrated in the northwest regions (Figure 2b,c).

### 3.5. The Relationship between Dietary Patterns and T2DM

A binary logistic regression model was employed to evaluate the association between three dietary patterns and the risk of T2DM. After adjusting for sociodemographic factors and behavioral information, no significant association was found between the oriental traditional pattern and the risk of T2DM (OR = 1.03, 95% CI: 0.93, 1.14). However, the highest-scoring group (Q4) of the animal-based pattern demonstrated a 1.15-fold increase in the risk of T2DM (OR = 1.15, 95% CI: 1.04, 1.27), whereas the highest-scoring group (Q4) of the plant-based pattern exhibited a 0.82-fold reduction in the risk (OR = 0.82, 95% CI: 0.74, 0.90) when compared to the lowest-scoring group (Q1) (Figure 3).

In this study, we conducted subgroup analyses based on gender, urban/rural residence, presence of chronic diseases, BMI, smoking, and alcohol consumption to further explore the association between dietary patterns and T2DM across diverse populations. The results of the subgroup analyses indicate that the findings of this study are robust. Specifically, the animal-based pattern appears to exert a stronger influence in males (*P* for interaction = 0.02), whereas the protective effect of the plant-based pattern is more prominent in females (*P* for interaction < 0.01). Furthermore, the oriental traditional pattern demonstrates an elevated risk of T2DM in urban dwellers (*P* for interaction = 0.01), and the protective effect of the plant-based pattern is more pronounced in non-smokers compared to smokers (*P* for interaction = 0.04) (Appendix A).

## 4. Discussion

### 4.1. Key Findings of the Study

In our analysis of a national sample comprising Chinese adults aged 45 and above, we have discerned notable regional dietary disparities within contemporary China. These disparities are delineated by distinct north–south demarcations, defined by the Yellow River and Yangtze River boundaries, and exhibit varying associations with T2DM. Within this study, we have identified three distinct dietary patterns: Plant-based Pattern, Oriental traditional Pattern, and Animal-based Pattern, which effectively encapsulate prevailing dietary habits.

The plant-based pattern predominantly prevails in the regions north of the Yellow River, while the oriental traditional pattern is distributed in the regions south of the Yangtze River. The animal-based pattern, on the other hand, is chiefly observed in the northwest region. It is worth noting that the animal-based pattern exhibits a positive association with T2DM risk, whereas the plant-based pattern demonstrates a negative association. Meanwhile, the oriental traditional pattern does not display a significant association with T2DM.

### 4.2. The Impact of Geographical Differences on Dietary Patterns

Geographical factors, encompassing climate, topography, and precipitation, wield substantial influence on human culture and profoundly shape the characteristics of different civilizations [16]. China’s distinctiveness lies in its two major rivers, the Yangtze and the Yellow River, each of which represents unique dietary cultural features [17]. The Yellow River basin signifies the northern expanse of China, characterized by an abundance of mountains, inadequate precipitation, and lower temperatures [18]. Additionally, the Yellow River is less amenable to transportation. Consequently, agriculture has historically been the predominant means of production in the northern region. This historical backdrop elucidates the predominance of plant-based pattern, primarily reliant on grains, north of the Yellow River within this study. In contrast, the southern region benefits from more favorable rainfall and temperature conditions, despite its intricate and diverse topography. The Yangtze River’s navigability has facilitated trade and transportation, engendering a wide array of available foods [19]. Hence, in our study, the southern region is typified by the oriental traditional pattern, which encompasses a rich variety of food types [20]. Lastly, the third dietary pattern elucidated in our study, the animal-based pattern, is predominantly concentrated in the northwest region. Historically, the northwest has been inhabited by nomadic communities, with animal husbandry serving as their principal means of sustenance [21]. Consequently, this region relies primarily on ample meat sources, resulting in a relatively monotonous dietary structure marked by limited consumption of fruits and vegetables. In this pattern, high-quality meat and dairy products constitute the primary sources of protein [22].

### 4.3. Association between Dietary Patterns and T2DM

In this study, the primary characteristics of the plant-based pattern encompass elevated consumption of whole grains, vegetables, legumes, and coarse cereals. This dietary pattern is characterized by its low fat and cholesterol content, low energy density, and a notable abundance of dietary fiber, collectively contributing to a reduced risk of T2DM. Consistent with previous research findings, plant-based dietary patterns exhibit a significantly negative association with T2DM [23,24,25], with legume consumption displaying particularly robust protective effects. Consumption of legumes can foster a healthier gut microbiota profile, enhance diversity, ameliorate cholesterol and lipid profiles, alleviate insulin resistance, and promote better blood glucose control [26,27,28]. These findings corroborate the outcomes of our study, where higher scores on the plant-based pattern were linked to lower FPG level.

The protective effect of plant-based pattern against T2DM may be attributed to its higher dietary fiber content. Dietary fiber mitigates elevated blood glucose level by retarding gastric emptying and reducing glucose absorption within the intestines, consequently alleviating insulin resistance [29]. Furthermore, increased fiber intake can lead to reduced consumption of high-energy foods, thereby mitigating the risk of overweight/obesity, a substantial T2DM risk factor [30]. Dietary fiber also stimulates the secretion of gastrointestinal hormones, such as cholecystokinin and glucagon-like peptide-1, thereby promoting insulin secretion [31].

Moreover, dietary fiber intake can enhance the proliferation of beneficial microbiota, including lactobacilli, bifidobacteria, and Akkermansia, thereby improving gut barrier function and lowering systemic inflammation level, including interleukin-1β, interleukin-6, monocyte chemoattractant protein-1, and tumor necrosis factor-alpha [32,33]. Consequently, dietary fiber plays a protective role in mitigating the risk of T2DM.

Contrarily, compelling evidence indicates an elevated risk of T2DM associated with red meat consumption [34], consistent with the findings of this study. Even after adjusting for covariates, the animal-based pattern remains significantly association to the risk of T2DM. Several plausible biological mechanisms can elucidate the relationship between red meat consumption and T2DM risk. Animal-based pattern exhibits a robust association with obesity, which in turn may heighten the T2DM risk [35]. Elevated red meat intake often accompanies increased saturated fat and cholesterol consumption, potentially fostering an energy surplus and augmenting the risk of abdominal obesity [36]. Moreover, red meat typically contains high levels of saturated fat and branched-chain amino acids, which can elevate serum-free fatty acid levels and induce insulin resistance in the liver and muscles [37]. Furthermore, the incorporation of nitrate–nitrite preservatives during the cooking process of processed meat products, coupled with elevated levels of heme iron, salt, saturated fatty acids, and the generation of advanced glycation end products, are often considered significant contributors to inflammation and oxidative stress, thereby accelerating insulin resistance and increasing the risk of T2DM [38,39,40,41]. In a cohort study conducted among the Rotterdam population, daily consumption of 50 g of processed meat was associated with a 0.12 mg/L increase in C-reactive protein level and an elevated risk of diabetes [42]. High heme iron intake was further linked to a 46% increased risk of diabetes over an average 8.8 years of follow-up in comparison to low heme iron intake [43].

Furthermore, in the subgroup analysis of this study, it was found that the animal-based pattern has a stronger effect in males, while the protective effect of the plant-based pattern is more pronounced in females. This explains the potential gender-specific preferences in dietary pattern selection, with males often having a higher frequency of meat consumption and a preference for animal-based foods, while females may tend to choose more vegetarian diets [44,45]. Additionally, smoking has consistently been identified as a significant risk factor for T2DM [46], which may explain the attenuation of the protective effect of the plant-based pattern in the smoking population in this study.

In this study, the oriental traditional pattern exhibits resemblances to the traditional Jiangnan dietary pattern in southern China and the Mediterranean dietary pattern. These patterns are characterized by a diverse array of foods, substantial daily vegetable consumption, and moderate fruit intake. Notably, the Mediterranean dietary pattern has long been acknowledged for its protective effects against T2DM [8]. However, a key distinction lies in the fact that both the traditional Jiangnan and Mediterranean patterns primarily rely on whole grains as their principal carbohydrate source [19]. In contrast, this study’s oriental traditional pattern predominantly features refined grains as the primary carbohydrate source. Elevated consumption of refined grains is recognized as a significant T2DM risk factor [47], potentially explaining the absence of an association between the oriental traditional pattern and T2DM risk in our study. The subgroup analysis results suggest that the oriental traditional pattern in urban populations is linked to a higher risk of T2DM, potentially stemming from greater consumption of refined grains among urban residents, thereby heightening the associated risk. Survey findings indicate lower whole grain intake among urban residents in China [48], which may consequently lead to increased consumption of refined grains, further emphasizing the risk of T2DM associated with refined grain consumption. The uneven technological development and industrial structures in China’s regional economy can result in varying levels of air pollution across different areas. Rapid urbanization and industrialization have led to declining environmental quality in the southern regions [49]. Additionally, the use of non-clean energy sources for cooking in these areas is linked to heightened exposure to PM2.5 [50], a significant risk factor for abnormal glucose metabolism [51], potentially impacting the relationship between oriental traditional pattern and T2DM. Therefore, this does not imply that the oriental traditional pattern identified in this study should be discouraged. Beyond the higher intake of refined grains, the oriental traditional pattern also encompasses increased consumption of vegetables and seafood, as well as beneficial nutrients such as polyunsaturated fatty acids, niacin, magnesium, and zinc, all of which are believed to exert a protective effect against T2DM [52,53,54,55]. Therefore, when adjusting the intake of whole grain and refined grain consumption, the oriental traditional pattern still holds substantial reference value.

### 4.4. Strengths and Limitations

This study exhibits several notable strengths. Firstly, the research data was obtained from a nationwide survey, enhancing its representativeness and thereby bolstering the credibility of the findings. Consequently, the conclusions drawn from this study bear significant implications for public health practices. This not only serves to underscore the pivotal roles of specific dietary components in the context of T2DM but also furnishes practical recommendations for future nutritional intervention programs. Secondly, we have visually portrayed the geographical distribution of three distinct dietary patterns across the nation. This visualization provides indispensable reference points for investigating regional and dietary variations and their impact on T2DM. Nonetheless, there are still several potential limitations to acknowledge. Firstly, the utilization of a cross-sectional design precludes the establishment of a causal relationship between dietary patterns and the risk of T2DM. However, cross-sectional studies do offer insight into individual exposures and outcomes at the time of the survey, thereby aiding in the identification of disease risk factors. Additionally, throughout the course of this study, we had to make subjective decisions. These decisions encompassed the categorization of foods into food groups, determination of the number of factors to extract, application of rotation methods, and labeling of factors. Moreover, diet is only one of the factors influencing T2DM, and other factors such as exercise and lifestyle, which were not fully taken into account in this study, may affect our analysis and conclusions. Finally, to ensure maximum inclusion of all T2DM patients, this study did not exclude individuals already diagnosed with T2DM and following dietary control measures. However, this decision may impact the assessment of the association between dietary patterns and T2DM.

## 5. Conclusions

In conclusion, our study reveals notable regional disparities in dietary patterns throughout China. Predominantly, a plant-based pattern prevails in the northern regions, while the southern regions adhere to an oriental traditional pattern, and the northwest leans towards an animal-based pattern. Notably, the animal-based pattern exhibits an increased risk of T2DM, in contrast to the plant-based pattern, which is associated to a reduced T2DM risk. For individuals diagnosed with T2DM, it is advisable to prioritize the consumption of whole grains and vegetables instead of meat. Furthermore, the characteristic of the diverse food offerings within the oriental traditional pattern retains its value. These findings provide further insights into comprehending the association between dietary patterns and the risk of T2DM.

## Figures and Tables

**Figure 1 nutrients-16-00107-f001:**
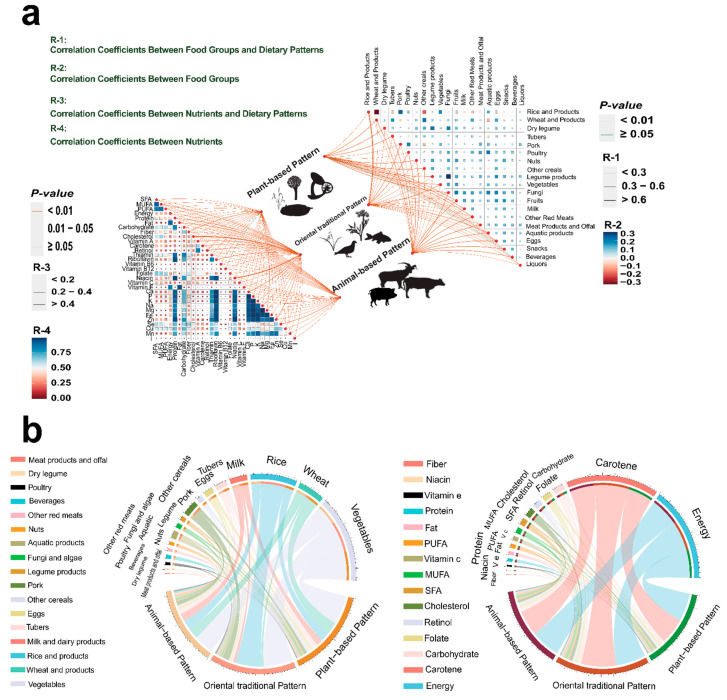
Characteristics of dietary patterns. Line colors indicate the significance (*p*-value) of the correlations between dietary patterns and food groups/nutrients, while line thickness represents the strength of these correlations (R-1, R-3). The triangular heat maps on both sides depict the internal correlations among the food groups and nutrients (R-2, R-4) (**a**). After categorizing the sscores into quartiles according to dietary patterns, (**b**) depicts the food groups and nutrient intake levels within the top quartile (Q4) of each dietary pattern. The width of the ribbon corresponds to the quantity of intake. In order to present the information as clearly as possible, selectively showcases the food groups and nutrients that exhibit significant differences in intake among the three dietary patterns. For comprehensive intake details, refer to Appendix A.

**Figure 2 nutrients-16-00107-f002:**
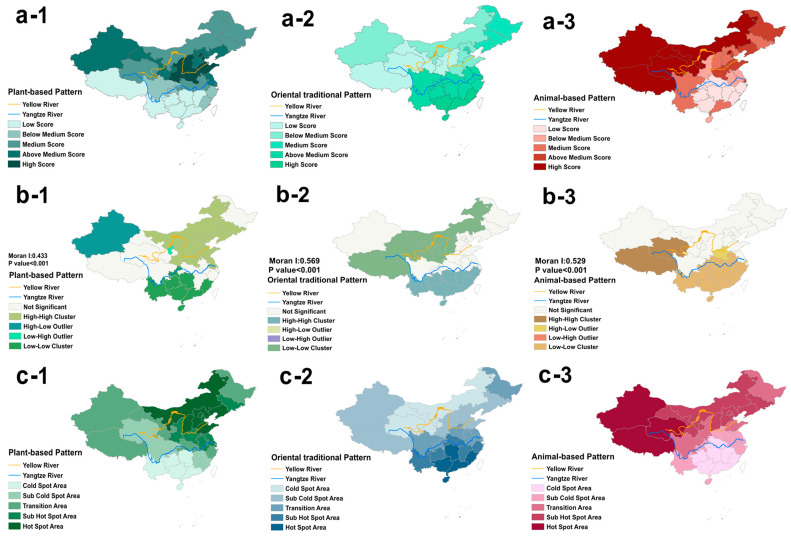
The spatial distribution of the three dietary patterns. (**a-1**–**a-3**) represent the actual distribution of three patterns, (**b-1**–**b-3**) represent the local spatial autocorrelation results of three patterns, and (**c-1**–**c-3**) represent the distribution of hot and cold spots of three dietary patterns. High-High Cluster: Regions with high dietary pattern scores clustering spatially with regions with high dietary pattern scores. Low-Low Cluster: Regions with low dietary pattern scores clustering spatially with regions with low dietary pattern scores. High-Low Outlier: Regions with high dietary pattern scores clustering spatially with regions with low dietary pattern scores. Low-High Outlier: Regions with low dietary pattern scores clustering spatially with regions with high dietary pattern scores. Hotspot area: Regions with statistically significant high dietary pattern scores that were surrounded by neighboring areas with similar high dietary pattern scores. Coldspot area: Regions with statistically significant low dietary pattern scores that were surrounded by neighboring areas with similar low dietary pattern scores.

**Figure 3 nutrients-16-00107-f003:**
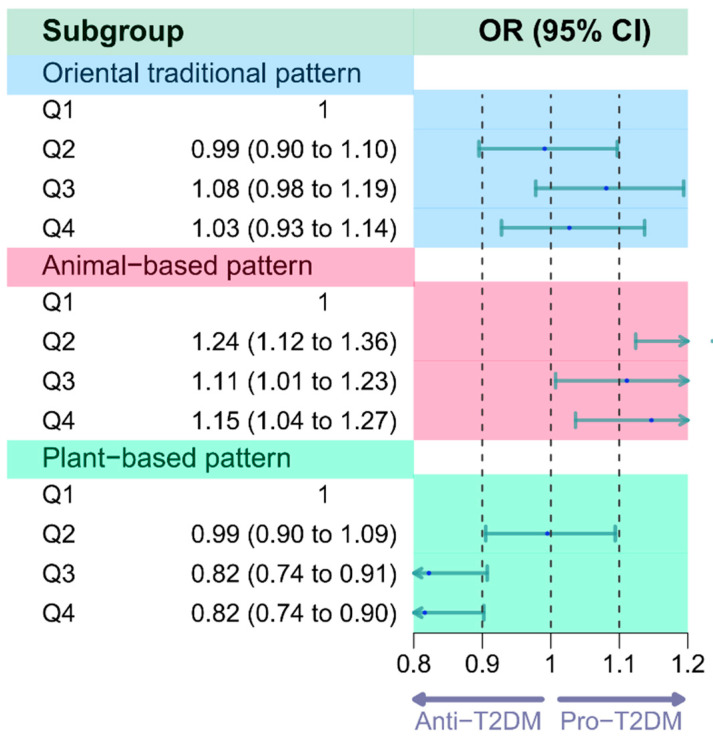
Logistic regression analysis on the association between the dietary patterns and T2DM. Adjusted for age, gender, income, educational level, marital status, region, ethnicity, chronic family history, BMI, physical activity, smoke, drinking, and total energy intake.

**Table 1 nutrients-16-00107-t001:** Characteristics of subjects according to T2DM status.

	Overall *N* = 36,648	Non-T2DM *N* = 32,665	T2DM *N* = 3983
Female (*n*, %)	19,982 (54.5)	17,680 (54.1)	2302 (57.8)
Age (Median, Q1, Q3,y)	58.6 (51.5; 65.0)	58.2 (51.2; 64.6)	61.4 (54.2; 67.2)
Age group (*n*, %)			
45–59 y	20,297 (55.4)	18,564 (56.8)	1733 (43.5)
60–74 y	14,602 (39.8)	12,579 (38.5)	2023 (50.8)
≥75 y	1749 (4.8)	1522 (4.7)	227 (5.7)
Rural (*n*, %)	21,988 (60.0)	20,113 (61.6)	1875 (47.1)
Income (*n*, %)			
<5000 Yuan/month	28,981 (79.1)	26,032 (79.7)	2949 (74.0)
5000–9999 Yuan/month	5939 (16.2)	5126 (15.7)	813 (20.4)
≥10,000 Yuan/month	1728 (4.7)	1507 (4.6)	221 (5.6)
Han ethnicity (*n*, %)	33,291 (90.8)	29,584 (90.6)	3707 (93.1)
Educational level (*n*, %)			
Below junior high school	20,797 (56.7)	18,667 (57.1)	2130 (53.5)
Junior high school	14,526 (39.7)	12,902 (39.5)	1624 (40.7)
Senior high school or above	1325 (3.6)	1096 (3.4)	229 (5.8)
Having a partner (*n*, %)	34,619 (94.5)	30,897 (94.6)	3722 (93.4)
Adequate physical activity (*n*, %)	32,743 (89.3)	29,344 (89.8)	3399 (85.3)
Smoking (*n*, %)	9711 (26.5)	8843 (27.1)	868 (21.8)
Drinking (*n*, %)	13,131 (35.8)	11,875 (36.4)	1256 (31.5)
BMI(Median, P25th, P75th, kg/m^2^)	24.1 (21.9;26.5)	23.9 (21.7;26.3)	25.7 (23.5;28.1)
BMI group (*n*, %)			
Underweight	1164 (3.2)	1119 (3.4)	45 (1.1)
Normal weight	16,686 (45.5)	15,521 (47.5)	1165 (29.2)
Overweight	13,588 (37.1)	11,852 (36.3)	1736 (43.6)
Obesity	5210 (14.2)	4173 (12.8)	1037 (26.1)
Family history of chronic diseases (*n*, %)	16,703 (45.6)	14,489 (44.4)	2214 (55.6)

## Data Availability

The data presented in this study are not allowed to disclose. The data are not publicly available due to [the copyright of the dataset in this study is currently owned by the Chinese Center for Disease Control and Prevention and has not been fully disclosed].

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
