# Peer review of "Geographical Distribution of Dietary Patterns and Their Association with T2DM in Chinese Adults Aged 45 y and Above: A Nationwide Cross-Sectional Study"

_nutrients, 2023, doi:10.3390/nu16010107_

Round 1

Reviewer 1 Report

Comments and Suggestions for Authors

The abstract overstates the  impact of the animal based pattern.  While risk is increased, saying "animal-based pattern carrying high risks or T2DM" does not seem to be warranted.

It is not clear what the 3 24-hour recalls are used for in the paper since the food frequency is used to develop the dietary patterns. 

The p values in table 1 are the result of the large sample size and are not useful consider omitting them. 

Figure 1 is too busy-- letters are hard to read and it's not clear from the figure what the reader should take from panel b.

In figure 3, the horizontal axis for OR1, OR2 and Or3 are on different scales making it confusing for the reviewer.  Use the same scale for all 3. 

Limitations do not include other factors that could contribute to the incidence of T2DM such as exercise.

Comments on the Quality of English Language

Some English language editing is required.

Author Response

Dear Esteemed Reviewer,

We are grateful for your valuable suggestions, and we have implemented the following modifications in response to your feedback:

  1. In consideration of your observation regarding the overstatement on the impact of the animal-based pattern in the abstract, we have revised it to state " animal-based pattern may increase the risk of T2DM."
  2. The role of the 24-hour food record in the paper was not adequately explained. This has been rectified in the methods section (2.4), wherein we have provided a detailed explanation of its purpose in evaluating the daily nutrient intake of the subjects.
  3. As per your suggestion, all p-values in Table 1 have been removed.
  4. Figure 1 has been redrawn with reduced content and color saturation to enhance clarity and conciseness. Additionally, footnotes have been added to aid the readers in retrieving information.
  5. Furthermore, Figure 3 has been redrawn to remove superfluous model information and ensure that all OR values are presented on the same scale on the horizontal axis.
  6. We duly acknowledge your comprehensive consideration of factors such as exercise and their potential impact on the incidence of T2DM. Therefore, we have added relevant discussion in the limitations section regarding the potential impact of factors like exercise on the study results.
  7. Finally, the language of the article is reviewed and modified by the native speaker. Thanks for your suggestion.

Reviewer 2 Report

Comments and Suggestions for Authors

Comments on the Quality of English Language

The manuscript needs some proof-reading for grammar and word choice.  

Author Response

Dear Esteemed Reviewer,

We appreciate your valuable suggestions, and the following modifications have been made accordingly.

1. “Spatial” was substituted with “Geographical” in the title, while dietary pattern or dietary model was usually used to describe the diet structure and components in many articles. Thus, we consider that it may be appropriate to keep it in the title and the whole manuscript.

2. In fact, the principal component analysis could not categorize the participants into mutually exclusive dietary patterns, each participant was scored in several obtained diet patterns through factor analysis, so those being in the highest quartile of a certain dietary pattern means that they are more inclined to that pattern. When compared with the lowest quartile of this dietary pattern, it showed the positive or negative association with the outcome.

3. As for the methods, in recognition of your valid point about the lack of comprehensive explanation of the principal component analysis, we have supplemented the details of the principal component method in the methods section (2.4) and explained the indicators of principal component analysis in the results section (3.2).

4. You pointed out that we did not explain how the dietary pattern scores were derived, which was indeed an oversight. We have addressed this in the methods section, explaining that the standardized scores generated after principal component analysis were used as the scores for each dietary pattern. Thank you for your suggestion.

5. You pointed out the meaningful work of adding interaction terms to the analysis. Following this advice, we have integrated the relevant subgroup analysis (Table S5) and provided corresponding explanations in the Results and Discussion sections. We appreciate your valuable suggestion, which has bolstered the rigor and scientific validity of the article.

6. As for your suggestion of discussion, we discussed the industrialization in the urban districts and added the corresponding reference evidence. Thanks for your advice.

7. Some minor points were revised after a careful reading of the full text. Thanks.

Reviewer 3 Report

Comments and Suggestions for Authors

This paper aims to investigate the spatial distribution of dietary patterns and their association with T2DM among Chinese adults aged 45 years and above.  On first read it appears to be interesting and worthwhile. The sample size is enormous, 36, 648, so any non-significant result would be unexpected.

The methodology is reasonably, sound, but the authors are scant on detail on numerous occasions.   Principal components analysis was used and some factor loadings in table S1 are provided.  What cutoff was used to hide the loading not reported?  How did you deal with cross loadings?  How did you arrive at your labels e.g., animal-based pattern, when nuts and fruit are part of this group  What are the fit statistics e.g., percentage of variance explained, communalities, eigenvalues to allow the reader to judge the usefulness of your model?

The geospatial techniques used are sophisticated and most readers will not be aware of them,  so the authors should devote a sentence or two to concepts such as Moran’s I in the statistical method section.  

Figure 1 is colourful, but difficult to interpret and understand what the numbers mean.  A footnote of some length should be added to explain exactly how to interpret the number in the columns.

Figure 2 is equally impenetrable.  Does a1, a2,  etc have some meaning or are they arbitrary label? What are the clusters that the authors are referring to and what does high-high etc mean? Again, a lengthy footnote needs to be included to help the reader understand.

The logistic regression table is interesting but most of the information is superfluous as the most reliable results are those provided in model 3.  Any differences from model 3 in models 1 or 2 are likely to be biased and confounded  Hence, models 1 and 2 should be deleted.  It’s not clear why the authors didn’t produce a  table showing all the associations for the other covariates in the model.

The upshot is that, whilst the paper is an interesting read, the authors need to devote some time and effort into helping and convincing a reader who is not an expert in their area that the results can be relied upon.

Author Response

Dear Esteemed Reviewer,

We appreciate your valuable suggestions, and the following modifications have been made accordingly.

1.You highlighted remaining deficiencies in the description of the principal component analysis method, which requires further supplementation. Following your suggestion, we have enhanced the description of the principal component method in the Methods section (2.4) and provided an explanation of the principal component analysis indicators in the Results section (3.2).

2.You pointed out deficiencies in our description of spatial statistics. Following your suggestion, we have furnished a more detailed explanation of the spatial analysis method in the Methods section (2.6) to enhance reader comprehension.

3.Your suggestion regarding Figure 1 is highly valuable. In response to your advice, we have added a detailed caption to explain the information in Figure 1.

4.You pointed out that Figure 2 also requires a detailed caption to explain the information. Following your suggestion, we have added a detailed caption to explain the information in Figure 2.

5.Following your advice, we have removed model1 and model 2 as they were deemed redundant, retaining only model 3. Additionally, we have added subgroup analysis to analyze the association with other covariates (Table S5).